# Topologically stable ergodicity breaking from emergent higher-form symmetries in generalized quantum loop models

Charles Stahl$^\star$, Rahul Nandkishore and Oliver Hart$^\dagger$

*Department of Physics and Center for Theory of Quantum Matter, University of Colorado Boulder, Boulder, Colorado 80309 USA*

(Dated: April 10, 2023)

$^\star$charles.stahl@colorado.edu, $^\dagger$oliver.hart-1@colorado.edu

We present a set of generalized quantum loop models which provably exhibit topologically stable ergodicity breaking. These results hold for both periodic and open boundary conditions, and derive from a one-form symmetry (notably not being restricted to sectors of extremal one-form charge). We identify simple models in which this one-form symmetry can be emergent, giving rise to the aforementioned ergodicity breaking as an exponentially long-lived prethermal phenomenon. We unveil a web of dualities that connects these models, in certain limits, to models that have previously been discussed in the literature. We also identify nonlocal conserved quantities in such models that correspond to a pattern of system-spanning domain walls, and which are robust to the addition of arbitrary $k$-local perturbations.

# 1   Introduction

When can many-body quantum systems fail to reach thermal equilibrium under their own dynamics, and thereby exhibit long-time behavior that lies fundamentally beyond equilibrium quantum statistical mechanics? The oldest answer to this question is "when they are integrable" [1]—integrable systems have extensively many explicit conservation laws, and do not thermalize to any conventional statistical mechanical ensemble. However, it is unclear to what extent (if at all) integrability is robust to generic perturbations, in the thermodynamic limit. A more recent answer is "when they are many-body localized" [2, 3]. Many-body localization by strong disorder involves extensively many emergent conservation laws, and does exhibit robustness to spatially local perturbations, although the proof of robustness is limited to one-dimensional spin chains [4], and moreover is subtle and has been questioned [5]. Quantum many-body scars [6–9] provide yet another example, without any conservation laws (but generically also without any notion of robustness). Still more recently, it was realized that the interplay of finitely many 'multipolar' conservation laws could break ergodicity [10], a result that was explained in terms of a shattering (aka fragmentation) of Hilbert space [11–13], whereby the unitary time evolution matrix block diagonalizes (within each symmetry sector) into exponentially many dynamically disconnected subsectors. This phenomenon has a simple proof of robustness [11] to arbitrary symmetric perturbations with strict spatial locality, and has stimulated a great deal of work into quantum dynamics with multipolar symmetries [14–34].

Very recently, a new route to ergodicity breaking was identified [35] which exhibits *topological* stability in two spatial dimensions. For the first time, the ergodicity breaking

is robust to spatially *nonlocal* perturbations, with the only requirement being that of *k-locality*, i.e., that the perturbation should act on no more than $k$ degrees of freedom (with $k/L \to 0$ in the thermodynamic limit, where $L$ is linear system size). This argument relied on a one-form symmetry, with the results being exact in the presence of arbitrary one-form-symmetric $k$-local perturbations. It was also explained how the one-form symmetry could be 'emergent' in a simple model of spin-1/2 degrees of freedom, which we hereafter refer to as the $\mathsf{CZ}_p$ model, in which case the ergodicity breaking became prethermal, and robust to arbitrary $k$-local perturbations up to exponentially long timescales. However, the results of Ref. [35] relied on a *dense packing of system winding loops*, such that the ergodicity breaking only arose in sectors of extremal one-form symmetry charge, and was 'all or nothing' (i.e., either the one-form symmetry sector was shattered into fully frozen one-dimensional subblocks with no dynamics, or it was not fragmented at all). Moreover, the construction presented in Ref. [35] was limited to systems with periodic boundary conditions.

In this work, we drastically extend the results of Ref. [35] to a much broader class of quantum loop models, without the requirement of either dense packing or periodic boundary conditions. We present a family of simple models with emergent one-form symmetry [36–40] (up to an exponentially long prethermal timescale). This family of models includes, as its simplest member, the $\mathsf{CZ}_p$ model of Ref. [35]. We show how this family of models generically exhibits ergodicity breaking with topological stability, and how the results may be extended to systems with open boundary conditions. Moreover, the broader family includes models in which the ergodicity breaking occurs in all symmetry sectors, and is not 'all or nothing' (i.e., symmetry sectors can block diagonalize into subblocks that are not one-dimensional). In this latter case, we identify certain nonlocal conserved quantities that robustly label the emergent subblocks. We also unveil a web of dualities that connects our results (in various limits) to higher-dimensional generalizations of several models previously considered in the literature, including the pair-flip model [41, 42], and which also make a connection with (quantum) square ice [43–45] and close-packed dimer models [46–53].

The manuscript is structured as follows: We start with a review of the one-dimensional pair-flip model in Sec. 2. This discussion serves as a warm-up and emphasizes the aspects of the model that will be most useful in our two-dimensional generalization. Readers familiar with the physics of the one-dimensional pair-flip model can therefore skip to Sec. 3, where we introduce a two-dimensional model that displays the aforementioned topological fragmentation. Sec. 3.1 contains a summary of the two-dimensional model that is meant to be entirely self-contained, so that readers may understand all the physics of the topological fragmentation. The rest of that section is devoted to details of the model, including its boundaries, symmetries, robustness, and fragmentation. In Sec. 4 we provide connections to square ice, the $\mathsf{CZ}_p$ model [35] (via a Kramers-Wannier-like duality), and a three-dimensional generalization. Finally, we discuss some open questions in Sec. 5.

## 2 Warm-up: One-dimensional pair-flip dynamics

We start with a brief review of the dynamics of the one-dimensional (1D) pair-flip (PF) model [41, 42], which exhibits some properties that will be useful for understanding our generalized quantum loop models. Readers familiar with the pair-flip model may safely skip this section and begin reading Sec. 3.

Consider a lattice composed of $L$ spin-$S$ degrees of freedom on the edges of a one-dimensional lattice. It will be convenient to work with the variable $m = 2S + 1$ in place of $S$; each degree of freedom is then associated to an $m$-dimensional local Hilbert space. Let us introduce the following graphical representation of states,

$$|0\rangle \equiv |\!\!\;\text{\textcolor{red}{$\bullet$}}\;\!\!\rangle \,, \quad |1\rangle \equiv |\!\!\;\text{\textcolor{green}{$\bullet$}}\;\!\!\rangle \,, \quad |2\rangle \equiv |\!\!\;\text{\textcolor{blue}{$\bullet$}}\;\!\!\rangle \,, \tag{1}$$

where we have set $m = 3$ for convenience of illustrations.

Given projectors $\hat{\mathcal{P}}_e^\alpha = |\alpha\rangle\langle\alpha|_e$ that project the spin at edge $e$ onto state $\alpha$, we may define the symmetry charges

$$\hat{N}^\alpha = \sum_e (-1)^e \hat{\mathcal{P}}_e^\alpha \,, \quad \alpha = 0, \dots, m-1 \,, \tag{2}$$

where $(-1)^e = 1$ if $e$ is in the even sublattice and $(-1)^e = -1$ if $e$ is in the odd sublattice. While it may appear that there are $m$ different generators in (2), only $m-1$ of them are independent since the projectors obey $\sum_{\alpha=0}^{m-1} \hat{\mathcal{P}}_e^\alpha = \mathbb{1}$ on every edge. The U(1)$^{m-1}$ symmetry (2) breaks the $m^L$-dimensional Hilbert space into $\mathcal{O}(L^{m-1})$ symmetry sectors. However, as we will show, following the discussion in Ref. [42], nearest-neighbor, symmetry-respecting dynamics is not fully ergodic within these sectors.

The most generic nearest-neighbor dynamics consistent with (2) is as follows: If the spins on edges $e$ and $e+1$ are both in state $\alpha$, flip them both to state $\beta$, with $0 \leq \alpha, \beta \leq m-1$. These "active" pairs can be represented by pairing up the legs emanating from spins into "dimers." For example,

$$|\!\!\;\text{\textcolor{red}{$\bullet$}}\;\text{\textcolor{red}{$\bullet$}}\;\!\!\rangle \rightarrow |\!\!\;\text{\textcolor{red}{$\bullet\!\!-\!\!\bullet$}}\;\!\!\rangle \,, \tag{3}$$

and with analogous notation for neighboring pairs of green and blue sites. Consider the following configurations:

$$|\!\!\;\text{\textcolor{red}{$\bullet\!\!-\!\!\bullet$}}\;\text{\textcolor{green}{$\bullet\!\!-\!\!\bullet$}}\;\!\!\rangle \,, \qquad |\!\!\;\text{\textcolor{green}{$\bullet$}}\;\text{\textcolor{red}{$\bullet\!\!-\!\!\bullet$}}\;\text{\textcolor{green}{$\bullet$}}\;\!\!\rangle \,. \tag{4}$$

The configuration on the left is evidently fully active. The configuration on the right is also fully active since the central active red pair can be permuted to green, allowing the green edge spins to become active. In this way, the two configurations are actually connected via local pair-flip moves. Indeed, since any contiguous region of $2n$ spins paired into $n$ noncrossing dimers can be connected via local moves to the all red state (say), *any* configuration of $n$ noncrossing dimers can be generated via pair-flip moves.

Suppose that the following procedure is performed: First, all neighboring pairs are grouped[1] according to Eq. (3). Next, all paired spins are removed from the string, and the pairing procedure is applied again to the remaining spins. This procedure is repeated until the there remains a configuration of unpaired spins that cannot be paired up without introducing crossings between dimers.

Having completed this procedure, observe that unpaired spins are able to move past paired spin configurations.[2] The simplest example of this being

$$\left| \phi \; \bullet\!\!-\!\!\bullet \right\rangle \leftrightarrow \left| \phi \; \bullet\!\!-\!\!\bullet \right\rangle \leftrightarrow \left| \bullet\!\!-\!\!\bullet \; \phi \right\rangle \leftrightarrow \left| \bullet\!\!-\!\!\bullet \; \phi \right\rangle . \tag{5}$$

However, since the color of the unpaired spin remains fixed, the ternary string corresponding to the unpaired dots (i.e., having removed all intervening paired spins) is conserved. As an explicit example, under the procedure just described,

$$\left| \phi \; \overset{\frown}{\bullet\!\!-\!\!\bullet} \; \phi \; \bullet\!\!-\!\!\bullet \right\rangle \mapsto \left| \phi \; \phi \right\rangle . \tag{6}$$

The reduced pattern shown on the right-hand side is conserved under pair-flip dynamics. Note that we are assuming open boundaries for the purposes of this discussion.

We call such a pattern of unpaired dots a *label*; spin configurations associated to different labels cannot be connected via pair-flip dynamics and therefore belong to dynamically disconnected sectors known as Krylov sectors [13]. If our system has an even (odd) number of spins then the label must have an even (odd) length, but can otherwise be any length between 0 and $L$. For open boundary conditions, the first color in the label is arbitrary and no color may match its neighbor, so there are $m(m-1)^{j-1}$ labels of length $j$, except for the trivial label ($j=0$) of which there is only one (and only exists if $L$ is even). In all, this allows for

$$\sum_{n=0}^{\lceil L/2 \rceil - 1} m(m-1)^{L-2n-1} + (1 \text{ if } L \text{ is even}) = \mathcal{O}[(m-1)^L] \tag{7}$$

Krylov sectors. See Appendix B for the sizes of the Krylov sectors, and for the slightly different counting of sectors in the presence of periodic boundaries. The exponential scaling of the number of Krylov sectors together with the polynomial scaling of the number of symmetry sectors means that the PF model must exhibit fragmentation [42]. In particular, Krylov sectors with labels of length $L$ each consist of one fully frozen state with no allowed dynamics.

Furthermore, fragmentation exists in generic symmetry sectors. Any symmetry sector has a minimal length $L_{\min}$ on which it exists. Take a representative spin pattern from that sector and an uncharged motif such as $\left| \phi \; \phi \; \phi \; \phi \; \phi \; \phi \right\rangle$. There are six motifs of size six (and no smaller motifs, as shown in Appendix B) but only four have a first

---

[1]While this grouping is not unique, the nonuniqueness does not affect the label that the procedure produces.

[2]The hopping process however preserves the sublattice of the unpaired spin. This leads to additional conservation laws for periodic boundaries and even $L$, described in Appendix B.

spin that does not match the last spin of our chosen pattern. Then for systems of size $L_{\min} + 6n$, we have $n$ slots into which we can independently place four compatible patterns, which gives a lower bound of $4^{(L-L_{\min})/6}$ different Krylov sectors within any fixed symmetry sector. A more detailed count of the number of Krylov sectors belonging to each symmetry sector is given in Appendix B, but the argument here is enough to show that fragmentation is present in generic symmetry sectors.

In the pair-flip model, and in most models in the literature, fragmentation depends strongly on strict locality. In our example, we can see this to be true by including next-nearest-neighbor dynamics. Any move that swaps the states of the spins on edges $e - 1$ and $e + 1$ is allowed by the symmetry (2). Under this dynamics, any two states within the same symmetry sector may be transformed into each other, regardless of their label as previously defined, completely melting the fragmentation. In the next section we will introduce a two-dimensional (2D) model with similar fragmentation properties, but where arbitrary $k$-local terms ($k < 2L$) may be introduced while preserving the fragmentation—the extra dimension (and attendant one-form symmetry) endows the fragmentation with topological stability.

# 3   The quad-flip model

We now define the quad-flip model, a generalization of the PF model to 2D. Reference [42] shows that the simplest generalization of PF dynamics to 2D, using a 0-form symmetry, results in fragmentation that is not robust to generic local perturbations. Reference [35] shows that 1-form symmetries can lead to robust fragmentation. This motivates our generalization of the PF model to 2D using a 1-form symmetry.

## 3.1   Summary

We will first present a general overview of how generalized 2D loop models can exhibit robust Hilbert space fragmentation without reference to any particular Hamiltonian realization. Explicit local Hamiltonians that give rise to the desired loop models prethermally, and a more technical discussion of their conventional symmetries and nonlocal conserved quantities, are presented from Sec. 3.3 onwards.

Consider an $L \times L$ square lattice in two spatial dimensions with spin-$S$ degrees of freedom on the edges of the lattice. As in Sec. 2, we introduce the variable $m = 2S + 1$ to parameterize the size of the local Hilbert space on each edge. The states on each edge are given the following graphical representation

$$|0\rangle \equiv |\,\textcolor{red}{\bullet}\,\rangle \ , \ |1\rangle \equiv |\,\textcolor{green}{\bullet}\,\rangle \ , \ |2\rangle \equiv |\,\textcolor{blue}{\bullet}\,\rangle \ , \ \ldots \tag{1}$$

where we have set $m = 3$ for convenience of illustrations.

We can define a 1-form symmetry in 2D analogous to the symmetry from Sec. 2. In general, $n$-form symmetries consist of operators defined on $(d - n)$-dimensional

manifolds [36–40]. For any path $\mathcal{C}$ on the lattice (with a definite starting edge), define the one-form symmetry charges

$$\hat{N}^\alpha_{\mathcal{C}} = \sum_{e_j \in \mathcal{C}} (-1)^j \hat{\mathcal{P}}^\alpha_{e_j}, \quad \alpha = 0, \dots, m-1, \tag{8}$$

where the edges $\{e_0, e_1, \dots\}$ in $\mathcal{C}$ are ordered so that $e_{j+1}$ follows $e_j$ when following the path. Changing the numbering on the edges may send $\hat{N}^\alpha_{\mathcal{C}}$ to $-\hat{N}^\alpha_{\mathcal{C}}$, but will otherwise leave the operator unchanged. For any $\mathcal{C}$ consisting of $|\mathcal{C}|$ edges, the operators are not all independent, obeying $\sum_{\alpha=0}^{m-1} \hat{N}^\alpha_{\mathcal{C}} = 0$ for $|\mathcal{C}|$ even or $\mathbb{1}$ for $|\mathcal{C}|$ odd. Any contractible path on the square lattice has $|\mathcal{C}|$ even.

Let us restrict to states that are source free[3] so that $\hat{N}^\alpha_{\partial \mathcal{R}} = 0$ for any region $\mathcal{R}$ of the lattice. In particular, the region may be chosen to be a single face, which only permits configurations of the form

$$\left| \vcenter{\hbox{\includegraphics{fig1}}} \right\rangle, \ \left| \vcenter{\hbox{\includegraphics{fig2}}} \right\rangle, \ \left| \vcenter{\hbox{\includegraphics{fig3}}} \right\rangle, \tag{9}$$

along with configurations related to these by interchanging colors. The graphical representation we have utilized makes it clear that spins of a given color must therefore form unbroken loops, and that loops of differing colors cannot intersect with one another. The allowed Hilbert space does not have a tensor-product structure, but is still exponential in system volume $L^2$. Consequently, there is room for a properly extensive entropy, and it makes sense to talk about thermalization or lack thereof. In Sec. 3.3 we show the constrained Hilbert space dimension grows as $\sim W_m^{L^2}$, with $W_m \gtrsim \max[(2m-1)/m, \sqrt{m}]$, in contrast to the $m^{2L^2}$ growth of the unconstrained Hilbert space.

The nontrivial operators are those defined on noncontractible paths. These operators measure the symmetry charges. We will explore the compatibility constraints between the charges, and therefore the number of symmetry sectors, in Sec. 3.4. For now it is enough to say that this symmetry breaks the constrained Hilbert space into $\mathcal{O}(L^{m-1})$ symmetry sectors, as in the pair-flip model. However, as in the pair-flip model, dynamics within the constrained Hilbert space is not fully ergodic within these sectors, and this time the ergodicity breaking is topologically robust.

We now examine generic $k$-local dynamics compatible with the local constraint (9). On a given face of the lattice, a minimum of two spins must be flipped to preserve the constraint. These two spins must have the same color, otherwise the constraint will be violated via the introduction of a forbidden loop crossing:

$$\left| \vcenter{\hbox{\includegraphics{fig4}}} \right\rangle \xrightarrow{\checkmark} \left| \vcenter{\hbox{\includegraphics{fig5}}} \right\rangle, \qquad \left| \vcenter{\hbox{\includegraphics{fig6}}} \right\rangle \xrightarrow{\times} \left| \vcenter{\hbox{\includegraphics{fig7}}} \right\rangle. \tag{10}$$

Applying the left-hand transition in (10) to the four faces around a vertex gives rise to the minimal update (i.e., acting on the fewest possible spins) to the system compatible

---

[3]Violations of this constraint are sources in a U(1) lattice gauge theory interpretation of this model for $m = 2$. We continue to use this language analogously for $m > 2$. Within the source-free subspace the 1-form symmetry is topological [54] (a.k.a. relativistic [55]).

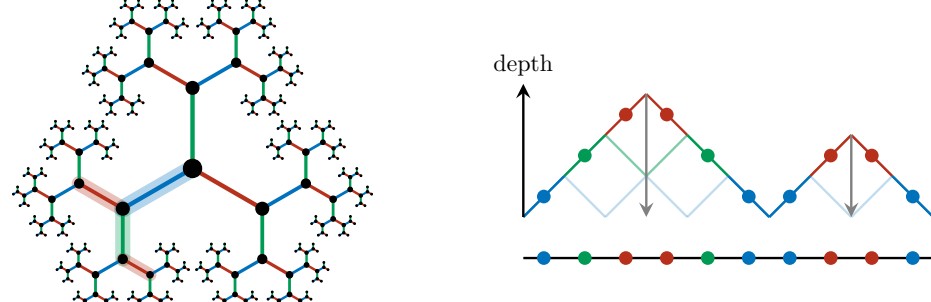

Figure 1: Left: Bethe lattice that can be used to define a height profile for the $m = 3$ loop model. Right: mapping of a one-dimensional cut of the square lattice to a colored Dyck path, where the height is determined by the depth of the corresponding position on the Bethe lattice. The corresponding edges have been highlighted on the Bethe lattice.

with the constraint: the four spins on the edges surrounding a vertex can simultaneously be flipped only if their colors match:

$$\left| \begin{array}{c} \end{array} \right\rangle \xrightarrow{\checkmark} \left| \begin{array}{c} \end{array} \right\rangle, \tag{11}$$

This update thereby modifies the color of the smallest closed loop configurations in the system, which are "active" in the same sense as the active pairs in 1D. This minimal update is a natural two-dimensional extension of the pair-flip dynamics discussed in Sec. 2. Hence, we refer to such updates as *quad-flip* (QF) dynamics.

QF dynamics can also change the color of any contractible loop. In fact, any two source-free states that differ from each other inside a contractible loop of a single color can be connected via a sequence of quad flips (11). To justify this statement, we map a given loop configuration to a height profile via the following mapping. We begin on a reference vertex of the square lattice, which corresponds to the root node of a Bethe lattice with coordination number $m$ (see Fig. 1 for an illustration). Then consider moving along an (arbitrary) real space path. Moving to a neighboring vertex on the square lattice implies traversing an edge of a particular color; on the Bethe lattice, this corresponds to hopping along an edge of the same color. Any sequence of colors encountered as we move along the real space path thus maps onto a particular sequence of moves on the Bethe lattice (e.g., if we encounter two red edges in succession in real space, then we hop along the red bond on the Bethe lattice and then back along that same red bond). Moreover, the local constraint (9) implies that the position on the Bethe lattice is independent of local deformations of the real space path.[4] A height profile can then be obtained using the (Hamming) distance to the root note on the

---

[4]This statement is only true in general for paths that differ by a closed, contractible path. If the height field is not single-valued upon winding around the system, this indicates the presence of nonlocal conserved quantities, as we will discuss.

Bethe lattice. This process is illustrated along a one-dimensional cut in Fig. 1. At a local maximum of the height field (depth $d > 1$), the height field must, by definition, decrease across all edges. Hence, at each such point, there exists a closed loop of a single color surrounding the local maximum. This loop can be flipped to be the same color as the loop at depth $d - 1$. This process can be repeated to connect *any* closed, noncrossing loop configuration contained within a contractible contour to a region of uniform color, so that the entire region becomes active. This also tells us that any local dynamics compatible with the 1-form symmetries can be reproduced by QF dynamics.

Once an active region wraps the system, it is surrounded by two noncontractible loops. There is no requirement that they are of the same color. See Fig. 2 for examples. If their colors differ, the colors of these winding loops cannot be modified via the previously described procedure. Instead, it may be necessary to simultaneously flip at least $2L$ spins to change the color of a noncontractible loop while remaining in the constrained subspace. Any noncontractible loop can still be deformed through the active regions arbitrarily, subject to the constraint (9).

Thus, QF dynamics naturally decomposes a 2D state into a collection of fluctuating closed loops that do not intersect, both contractible and noncontractible. The contractible loops can change color while the noncontractible loops cannot change color but can fluctuate past the contractible loops. Recall that in 1D the active pairs can change color while the unpaired spins cannot change color but can hop past the active pairs. This motivates a labeling procedure for QF dynamics similar to the procedure in 1D. First, choose a path that wraps the system once in the horizontal direction. Treat the pattern of spins along this path as a 1D system and use the procedure described in Sec. 2 to extract a label. This is the horizontal label, and there are $\mathcal{O}[(m-1)^L]$ such labels, as in 1D. Fig. 2 illustrates this procedure (with the lattice not drawn). Furthermore, deforming the path only inserts or removes matching pairs into the spin pattern, which are removed when the pattern is converted into a label. This means that topologically equivalent paths will result in the same label. Now, choose a path that wraps the system once in the vertical direction and extract a vertical label. The vertical and horizontal labels must satisfy some compatibility, discussed in Sec. 3.4, but the fact that the number of Krylov sectors grows exponentially in $L$ diagnoses fragmentation in the QF model.[5]

Finally, we can see that this fragmentation is *topologically robust*. The (exponentially numerous) irreducible labels for the Krylov sectors depend only on the sequence of *noncontractible* loops (of nonrepeating color), and noncontractible loops can neither change color nor fluctuate past noncontractible loops of a different color, unless we act on $\mathcal{O}(L)$ degrees of freedom. Thus, any $k$-local dynamics, with $k/L \to 0$, cannot change the Krylov sector label.

---

[5]Reference [42] defines fragmentation in 2D as the existence of $\exp(L^2)$ Krylov sectors and classifies $\exp(L)$ Krylov sectors as a subsystem symmetry. The QF model does not posses a subsystem symmetry but still has $\exp(L)$ Krylov sectors in each symmetry sector. The presence of nonlocal conserved quantities in the QF model justifies our use of the term fragmentation.

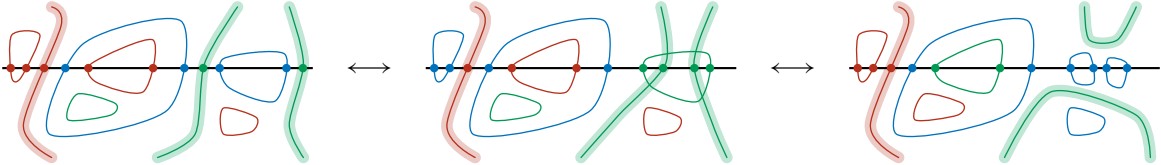

Figure 2: Schematic illustration of a valid configuration of noncrossing, contractible loops, and system-spanning loops (thick lines). Observe that a noncrossing dimer configuration is automatically created on the one-dimensional slice (black line) for contractible loops. Two adjacent noncontractible loops of the same color can be made contractible since loops of the same color are not forbidden from intersecting. In all configurations, only the red noncontractible loop contributes to the irreducible label.

## 3.2   Boundary conditions

Our discussion here will focus on open boundary conditions; discussion of periodic boundaries is relegated to Appendix B. It will be helpful to have two types of open boundaries: one type on which the symmetry operators may end and one on which they may not, similar in spirit to surface codes [56]. In addition to being more physically realizable, this will allow us to avoid the constraints between horizontal and vertical labels, so that the labeling system is more like the 1D version. The first boundary is the "smooth" boundary, with no edges sticking out. On this boundary we continue to define the symmetry to act on closed paths on the lattice that do not end. A source-free configuration of such a boundary may look like:

$$\text{（図）} \tag{12}$$

where all boundary faces are source free. Observe that loops are allowed to end on this boundary.

To obtain a system with nontrivial 1-form symmetry charges, we must find a boundary on which the symmetry operators can terminate. Ordinarily, this type of boundary, the "rough" boundary, has boundary edges sticking out of it. We find it more natural to define the rough boundary in the QF model to have every other boundary edge sticking out, so that all symmetry operators act on an even number of legs. Then, we define symmetry operators on paths $\mathcal{C}$ that terminate on the boundary edges. Source-free configurations, with $\hat{N}_{\mathcal{C}}^{\alpha} = 0$, are

$$\text{（図）} \tag{13}$$

and configurations related to these by interchanging colors, as in (9). This preserves the graphical constraint that loops of different colors may not intersect. An example

configuration is

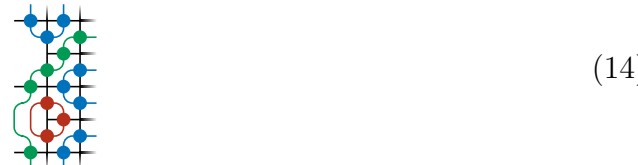

$$(14)$$

Note that loops cannot terminate on the rough boundary. Just as in the bulk, two symmetry operators that differ across a source-free region agree, so that the endpoints of a symmetry operator may be moved without changing its value:

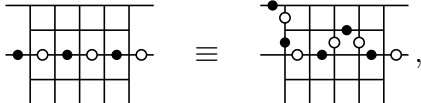

where closed and open circles distinguish between the two sublattices. If, however, we tried to define symmetry operators that end on the smooth boundary there is no reason they should give the same symmetry charges, so symmetry operators that stretch from the left to the right boundary are noncontractible.

Having defined the operators that measure the symmetry charge in (8), we should also define the operators charged under the symmetry. These are operators that locally commute with the constraint but may fail to commute with the nontrivial symmetry operators. To write such operators we must first choose a closed path $\mathcal{C}'$ on the dual lattice formed by placing vertices on the plaquette centers of the primary lattice. This path is required to turn at every vertex of the dual lattice. Then, the charged operators are

$$\hat{\mathcal{O}}_{\mathcal{C}'}^{\alpha\beta} = \prod_{i \in \mathcal{C}'} |\alpha\rangle\langle\beta|_i \,, \quad \alpha, \beta = 0, \ldots, m-1, \quad \alpha \neq \beta \,, \qquad (15)$$

which simultaneously projects onto dual paths where all edges are in the state $|\beta\rangle$ and then flips them all to $|\alpha\rangle$. These operators always commute with the constraint but act trivially if the edges in the chosen dual path are not all in the initial state $|\beta\rangle$. If $\mathcal{C}'$ wraps the system in the vertical (horizontal) direction, it may change the value of $\hat{N}_{\mathcal{C}}^{\alpha}$ if $\mathcal{C}$ wraps the system in the horizontal (vertical) direction.

In the bulk, the smallest possible charged operators correspond precisely to the quad flip (11). As discussed in Sec. 3.1, any contractible charged operator can be decomposed into a series of quad flips. The charged operators are allowed to end on the smooth boundary, in the sense that this can be consistent with the symmetry. In particular, a charged operator that acts on the three edges around a vertex may flip those three edges if they match. For example, the three blue edges in (12) may be flipped to either green or red.

Near the rough boundary, vertices have either three or four edges sticking out from them. The four-edge vertices behave like bulk vertices and support QF moves. The three-vertex edges behave like those on the smooth boundary and support triple flips. For example, the three red edges in (14) may be flipped to either green or blue. Both of

these scenarios correspond to closed loops that do not end on this boundary. In fact, no charged operators may end on the rough boundary.

A system with smooth boundaries on the top and bottom and rough boundaries on the left and right will have a nontrivial 1-form symmetry evaluated on horizontal system-spanning paths. The symmetry counts the number of nontrivial vertical system-spanning loops, whose endpoints can be shifted along the smooth top and bottom boundaries by the three-site flips at the smooth boundaries. Remember that loops cannot end on the rough (left and right) boundaries. Thus, our symmetry sectors for an $L \times L$ system are precisely those of the PF model on a length-$L$ system, but with a 2D 1-form symmetry instead of a 1D 0-form symmetry.

## 3.3   Quad-flip model and robustness

We are now ready to write down a family of Hamiltonians that realize QF dynamics in a full tensor-product Hilbert space, where the hard constraint is made soft, i.e., it is enforced energetically. To this end, we introduce a natural "parent" Hamiltonian,

$$\hat{H} = J \sum_f \sum_{\alpha=0}^{m-1} \left( \hat{N}_{\partial f}^{\alpha} \right)^2 - g \sum_e \sum_{\alpha=0}^{m-1} \sum_{\beta=0}^{m-1} \xi_e^{\alpha\beta} \, |\alpha\rangle\langle\beta|_e \, , \qquad (16)$$

where $J > 0$, $\hat{N}_{\partial f}^{\alpha}$ is defined in Eq. (8), the first sum is over the elementary faces $f$ of the square lattice, and the second sum runs over all edges $e$. The dimensionless coefficients $\xi_e^{\alpha\beta}$ are of order one and serve to break all discrete symmetries. We will proceed by treating the second term perturbatively within the groundspace of the first.

The first term is positive semidefinite and is minimized by spin configurations satisfying $N_{\partial f}^{\alpha} = 0$ on all faces $f$ (source-free configurations). As previously discussed, the source-free space has a U(1)$^{m-1}$ one form symmetry and grows as $W_m^{L^2}$. A Pauling estimate for $W_m$ may be obtained as follows. The total Hilbert space has size $m^{2L^2}$, since spins live on edges of a square lattice. If we pick a reference plaquette and go around clockwise, we have $m$ choices for the value of the first spin. The second spin can either match (one choice) in which case we have $m$ choices for the remaining two spins ($m^2$ total), or it can be different ($m - 1$ choices), in which case the remaining two spins are fixed [so $m(m - 1)$ choices total]. Altogether this gives $2m^2 - m$ satisfying assignments out of $m^4$, so a fraction $(2m-1)/m^3$ of each plaquette Hilbert space satisfies the constraint. There are $L^2$ plaquettes in all, so (treating plaquettes as independent) a fraction $[(2m - 1)/m^3]^{L^2}$ of the total Hilbert space satisfies the constraints, yielding a constrained Hilbert space of size $[(2m - 1)/m^3]^{L^2} \times m^{2L^2} \sim [(2m - 1)/m]^{L^2}$. This yields an estimate $W_m \approx \frac{2m-1}{m}$. For $m = 2$, this estimate is within 3% of the exact value of $W_2 = (4/3)^{3/2} \simeq 1.54$, derived in Sec 4.1 from a mapping to square ice. On the other hand, for large $m$, this estimate undercounts the number of source-free states. Placing elementary length-four loops around every other vertex gives $W_m = \sqrt{m}$ in the large-$m$ limit.[6] The important point is that the subspace is exponential in system

---

[6]We thank Ethan Lake for this observation.

volume so there is room for a nonzero entropy density, and it makes sense to talk about thermalization or lack thereof.

The second term is the most general single-site Hamiltonian, and will generate generic longer-range terms within perturbation theory.[7] Hermiticity requires that the matrix elements satisfy $\xi_e^{\alpha\beta} = \bar{\xi}_e^{\beta\alpha}$. We can view the matrix elements as creation and annihilation operators for sources/sinks, breaking the 1-form symmetry and leading to nontrivial mixing of the classical ground states in the eigenstates of Eq. (16). However, we will now show that, even in the presence of the off-diagonal matrix elements, QF dynamics are preserved up to order system size in perturbation theory.

Let us consider the lowest-order dynamics produced by the off-diagonal matrix elements within the constrained space. This is given by the term

$$\hat{H}_{\mathrm{QF}} = -\frac{g^4}{J^3} \sum_v \sum_{\alpha\neq\beta} \sum_{\beta=0}^{m-1} \xi_v^{\alpha\beta} \hat{A}_v^{\alpha\beta}, \qquad \xi_v^{\alpha\beta} = \frac{5}{16} \prod_{e\in v} \xi_e^{\alpha\beta} \tag{17}$$

where $\hat{A}_v^{\alpha\beta} = \prod_{e\in v} |\alpha\rangle\langle\beta|_e$ flips the four spins around vertex $v$ to state $|\alpha\rangle$ if they are all initially in state $|\beta\rangle$. We have omitted diagonal transitions, which lead to a trivial energy shift. Thus we recover QF dynamics, so we call (17) the QF Hamiltonian. The $5/16$ prefactor comes from summing all $4!$ ways of flipping the spins around a vertex at fourth order in perturbation theory.

For vertices $v_0$ on the smooth boundary, the lowest-order terms come at third order in perturbation theory. They have the form

$$\hat{H}_{\mathrm{QF}} = -\frac{g^3}{J^2} \sum_{v_0} \sum_{\alpha\neq\beta} \sum_{\beta=0}^{m-1} \xi_{v_0}^{\alpha\beta} \hat{A}_{v_0}^{\alpha\beta}, \qquad \xi_{v_0}^{\alpha\beta} = \prod_{e\in v_0} \xi_e^{\alpha\beta} \tag{18}$$

and $\hat{A}_{v_0}^{\alpha\beta}$ now acts on only three spins. On the rough boundary, perturbation theory generates $\hat{A}_v^{\alpha\beta}$ terms on three-edge vertices at third order and on four-edge vertices at fourth order.

We should recognize the bulk and boundary terms as the minimal charged operators around every vertex. Thus, we have entirely recovered the dynamics we considered in Sec. 3.1. If we consider higher orders in perturbation theory, we will generate new terms, but they will be decomposable into series of quad flips, as shown in Sec. 3.1. The result is that the fragmentation remains exact at *any* order in perturbation theory, up to order system size. At $\mathcal{O}(L)$ in perturbation theory, however, we can flip the color of a noncontractible loop, changing the one-form symmetry charge and removing the Krylov structure associated with the irreducible labels.

Although perturbative dynamics will not melt the Krylov sectors, there may be nonperturbative effects that do. At times exponentially long in $J/g$ the system is able to

---

[7]Since the basic structure of our dynamics follows purely from 1-form symmetries (which are emergent within the groundspace of the first term), it follows that the basic results will be robust to inclusion of arbitrary $k$-body perturbations, without requirement of spatial locality, as long as $k/L \to 0$ in the thermodynamic limit.

violate the source-free constraint and leave the groundspace of the classical Hamiltonian. Then, a noncontractible loop may break apart and retract. This exponentially long timescale is the prethermal timescale up to which the parent Hamiltonian realizes dynamics with an emergent one-form symmetry, and may be bounded using now standard techniques [57]. Imposing the source-free constraint exactly (instead of softly) amounts to setting the prethermal timescale for $k$-local dynamics to infinity, in the thermodynamic limit.

## 3.4   Fragmentation

We are now prepared to describe the fragmentation of the QF Hamiltonian in full detail. First, let us use the boundary conditions from Sec. 3.2, with smooth boundaries on the top and bottom and rough boundaries on the left and right. The groundspace has an emergent $U(1)^{m-1}$ one-form symmetry, defined on horizontal system-spanning paths. There are $\mathcal{O}(L)$ possible symmetry charges for each independent one-form symmetry, so overall there are $\mathcal{O}(L^{m-1})$ symmetry sectors taking the emergent one-form symmetry into account. The dynamics must, at minimum, be block diagonal by one-form symmetry sector. However, as we shall now show, each symmetry sector will further be fragmented into $\exp(L)$ dynamically disconnected subblocks.

The basic argument follows the discussion in Sec. 2 for the one-dimensional pair-flip model. Namely, we can take any horizontal system-spanning path on the lattice and consider the sequence of colors encountered along this effective one-dimensional system. We can then sequentially delete same-color neighbors until we are left with an irreducible one-dimensional sequence or 'label' of length $0 \leq \ell \leq L$ in which no two adjacent entries are the same color. It is straightforward to see that local deformations of the real-space path within the source-free subspace do not alter this irreducible label. For example, any system-spanning path terminating on the left and right boundaries of Fig. 3(a) returns the length $\ell = 2$ label $\left| \textcolor{green}{\blacklozenge} \ \textcolor{red}{\blacklozenge} \right\rangle$. This irreducible label essentially enumerates the sequence of distinct vertical system-spanning loops of nonrepeating color. We do not count adjacent loops of the same (i.e., repeating) color since these can locally reconnect and flip. In contrast to the one-dimensional PF model, in the two-dimensional QF model this irreducible label exhibits topological stability – the only way to change the irreducible label is to flip an entire system-spanning vertical loop, which requires acting on $\mathcal{O}(L)$ degrees of freedom. Thus, this irreducible label is conserved under any $k$-local dynamics (with $k/L \to 0$), and thus defines a topologically robust Krylov subsector.

The total number of symmetry sectors is $\mathcal{O}(L^{m-1})$, but, as discussed in Sec. 2, the total number of Krylov subsectors (now topologically robust) is $\mathcal{O}[(m-1)^L]$. This follows because labels can have length up to $L$ [see Fig. 3(b) for an example of a state with a label of length $\ell = L$], and once the first entry in the label is fixed each subsequent entry has $(m-1)$ choices. Thus, there are only polynomially many distinct symmetry sectors, but there are exponentially many dynamically distinct Krylov subsectors. It follows that the dynamics must exhibit fragmentation, and cannot be fully ergodic. It further follows that for $m \geq 3$ fragmentation arises in *every* symmetry sector. This

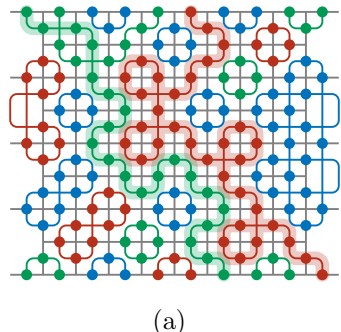
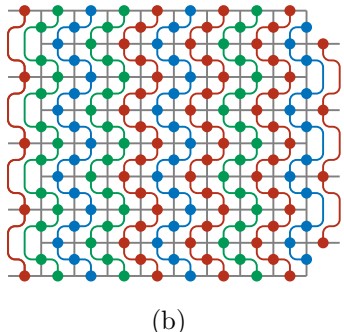

(a)                                                          (b)

Figure 3: (a) A state belonging belonging to the Krylov sector specified by the label $|\,\blacklozenge\ \blacklozenge\,\rangle$, which captures the presence of the green and red system-spanning loops (thick green and red lines, respectively). (b) A close-packed configuration of system-spanning loops that remains frozen under local dynamics. Note that the offset of the rough boundary differs on the left and right edges to accommodate a dense packing of loops.

follows by analogy to Sec. 2 because for any given value of the symmetry charge, there is a minimal system size $L_{\min}$ required to realize it. There exist 'charge-neutral' motifs with irreducible labels that can be 'glued' on, e.g., for $m = 3$ we can consider a sequence of vertical system-spanning loops of the form $|\,\blacklozenge\ \blacklozenge\ \blacklozenge\ \blacklozenge\ \blacklozenge\ \blacklozenge\,\rangle$, or five other patterns obtained from this one by permuting colors. Of the six motifs, four have a first spin that does not match the last spin of our chosen pattern. Gluing such a motif onto the system of size $L_{\min}$ does not change the symmetry charge but does multiply the number of Krylov subsectors by four. Such a process can then be iterated (bearing in mind that for $m = 3$ only four of the 'charge-neutral, gluable motifs' will have the property that the first loop of the motif does not have the same color as the last loop of the previous motif), giving a number of Krylov sectors that scales as at least $\sim 4^{(L-L_{\min})/6}$, for $m \geq 3$. Thus, the QF Hamiltonian (with $m \geq 3$) exhibits fragmentation in generic symmetry sectors, with the fragmentation furthermore exhibiting topological stability, being robust to arbitrary $k$-local perturbations as long as $k/L \to 0$ in the thermodynamic limit.

A special role is played by densely packed configurations, which generate labels of length $L$. As long as we stagger the rough boundary edges, as in Fig. 3, these configurations correspond to dense packings of noncontractible loops of nonrepeating color, and generate frozen states that have no dynamics under arbitrary $k$-local perturbations. For an example of a frozen state, see Fig. 3(b). The number of such configurations depends strongly on both the local Hilbert space dimension $m$ and on whether (and, if so, how) the lattice is terminated. With periodic boundary conditions, the pattern in Fig. 3(b) appearing in the top row can be shifted either left or right in each subsequent row, subject to the constraint that the pattern joins up around the periodic boundaries. With open boundary conditions, this freedom is not present, since the packing would cease to be dense at the left and right boundaries. For $m = 2$ there are an $\mathcal{O}(L^0)$ number of frozen configuration with open boundaries, although there are an $\exp(L)$

number of frozen configurations with periodic boundaries [35]. For $m \geq 3$, the number of frozen configurations grows exponentially with $L$ even for open boundaries; there are $m$ choices for color of the boundary loop, and $m - 1$ choices for each of the other loops. Hence, the number of frozen configurations is $\propto (m - 1)^L$. Exact (asymptotic) countings of the number of Krylov sectors and the number of frozen states are derived in Appendix B, for both open and periodic boundary conditions. With open boundaries we have $\mathcal{O}[(m - 1)^L]$ frozen states and $\mathcal{O}[(m - 1)^L]$ nonfrozen Krylov sectors for $m \geq 3$. With periodic boundaries we have $\mathcal{O}[(m - 1)^L]$ frozen states but only $\mathcal{O}[(m - 1)^L/L]$ nonfrozen Krylov sectors for $m \geq 3$.

We can also define the commutant algebra [42] for the QF model. The operators in the algebra generalize the symmetry operators (8) and are entirely analogous to those in the PF model, which were found in Ref. [42]. With our choice of open boundary conditions, choose a horizontal system-spanning path $\mathcal{C} = \{e_0, e_1, \ldots, e_K\}$ on the lattice, where $K \geq L$ is the length of the path. The nonlocal integrals of motion (IoMs) are

$$\hat{N}^{\alpha_1 \alpha_2} = \sum_{j_1 < j_2} (-1)^{j_1 + j_2} \hat{\mathcal{P}}_{j_1}^{\alpha_1} \hat{\mathcal{P}}_{j_2}^{\alpha_2}, \quad \alpha_1 \neq \alpha_2, \quad 0 \leq \alpha_1, \alpha_2 \leq m - 1$$

and the larger operators

$$\hat{N}^{\alpha_1 \alpha_2 \cdots \alpha_k} = \sum_{j_1 < j_2 < \cdots < j_k} (-1)^{\sum_l j_l} \hat{\mathcal{P}}_{j_1}^{\alpha_1} \hat{\mathcal{P}}_{j_2}^{\alpha_2} \cdots \hat{\mathcal{P}}_{j_k}^{\alpha_k}, \tag{19}$$

with[8] $0 \leq k \leq L$ and $\alpha_j \neq \alpha_{j+1}$. Within the source-free subspace the value of these operators does not depend on smooth variations of the choice of $\mathcal{C}$. As described in Ref. [42], in a system with $L$ even (odd), the IoMs with $k$ even (odd) are all linearly independent but the IoMs with $k$ odd (even) can be written in terms of the former. Just as we can view the symmetry operators as distinguishing symmetry sectors which consist of collections of Krylov sectors, we can view the IoMs as distinguishing successively more fine-grained sectors.

## 4   Generalized no-crossing models

Having introduced the symmetries that give rise to QF dynamics and its numerous conserved patterns, we move to discussing related models that exhibit analogous phenomena. These include models to which the QF Hamiltonian reduces in certain limits, as well as various dualities of the model, and its extension to higher dimensions. This section is not 'load bearing' for our basic story, and may be safely skipped by readers uninterested in connections to known models or dualities thereof.

To facilitate the presentation of the various dualities we discuss, it will be useful to define the following clock operators on each edge $e$, which act on the $m$-dimensional

---

[8]We could define similar operators for $k \leq K$, but these longer operators would contain redundant information.

Hilbert space introduced in Eq. (1)

$$\hat{\mathcal{Z}} \ket{\alpha} = e^{2\pi i \alpha/m} \ket{\alpha} , \quad \hat{\mathcal{X}} \ket{\alpha} = \ket{\alpha + 1} , \tag{20}$$

so that $\hat{\mathcal{X}}$ and $\hat{\mathcal{Z}}$ are unitary and obey $\hat{\mathcal{Z}}\hat{\mathcal{X}} = e^{2\pi i/m}\hat{\mathcal{X}}\hat{\mathcal{Z}}$, which generalizes the anti-commutation of Pauli matrices ($m = 2$). Since we identify $\ket{\alpha + m} \equiv \ket{\alpha}$, we also have $\hat{\mathcal{Z}}^m = \hat{\mathcal{X}}^m = \mathbb{1}$.

## 4.1   Square ice

For spin-$1/2$ degrees of freedom (i.e., $m = 2$), the parent Hamiltonian for the QF model that we present in Eq. (16) is unitarily equivalent to square ice [43–45] in a magnetic field, allowing us to make exact statements about the number of constraint-satisfying states. To see this, consider the unitary transformation that flips the sign of $\hat{\mathcal{Z}}$ operators on vertically oriented edges:

$$\hat{U} = \prod_{e : \text{vertical}} \hat{\mathcal{X}}_e . \tag{21}$$

Under this transformation, the constraint on faces that selects local ground state configurations becomes

$$\hat{N}_{\partial f}^{\uparrow} = \sum_{e_j \in \partial f} (-1)^j \hat{\mathcal{P}}_{e_j}^{\uparrow} \xmapsto{\hat{U}} \sum_{e_j \in \partial f} \hat{\mathcal{Z}}_{e_j} = 0 , \tag{22}$$

with the same expression for $\hat{N}_{\partial f}^{\downarrow}$, up to a sign. Hence, the constraint in this rotated basis corresponds to requiring that there exist two up spins and two down spins around every face of the square lattice, which (up to a sublattice-dependent sign) is equivalent to the two-in-two-out "ice rule" [58, 59]. For consistency with established conventions for square ice, we will interchange the vertices and faces of the square lattice, such that the constraints are on vertices. On one sublattice the six constraint-satisfying spin configurations take the form

$$\left| \begin{array}{c} \text{⊞} \end{array} \right\rangle \mapsto \left| \begin{array}{c} \text{-○⦁○-} \end{array} \right\rangle \equiv \left| \begin{array}{c} \text{→⦁←} \end{array} \right\rangle, \quad \left| \begin{array}{c} \text{⊞} \end{array} \right\rangle \mapsto \left| \begin{array}{c} \text{-○⦁⦁} \end{array} \right\rangle \equiv \left| \begin{array}{c} \text{→⦁→} \end{array} \right\rangle, \tag{23}$$

with the sign convention for the arrows reversed for vertices belonging to the other sublattice. The correspondence in (23) means that we can deduce the size of the constrained Hilbert space for the $m = 2$ QF model, since the number of two-in-two-out configurations is known to scale as $W_2^{L^2}$ with $W_2 = (4/3)^{3/2}$ [43, 60].

The QF dynamics described in Sec. 3 – in which closed loops of a single color passing through four spins are simultaneously flipped to a different color – can now viewed as "ring-exchange" dynamics. In particular, in the rotated basis, spins around a face can be flipped between clockwise and anticlockwise orientations:

$$\left| \begin{array}{c} \circlearrowright \end{array} \right\rangle \leftrightarrow \left| \begin{array}{c} \circlearrowleft \end{array} \right\rangle . \tag{24}$$

Since the states in Eq. (23) are in one-to-one correspondence with one another, ring-exchange dynamics acting within the manifold of two-in-two-out states will exhibit conservation laws identical to those in the $m = 2$ QF model. However, in the QF model with just two colors, fragmentation is confined to symmetry sectors of maximal one-form charge [35]; in all other sectors each symmetry sectors hosts just a single label. This is in stark contrast to the $m > 2$ generalizations discussed in Sec. 3, which exhibit fragmentation in *generic* symmetry sectors.

## 4.2 Dualities and generalized PXP models

### 4.2.1 From pair-flip to constrained spin flips

We now show how the pair-flip model in one dimension is dual to a parity-sensitive PXP-type model. We will work with $m = 3$ for simplicity, with the generalization to $m > 3$ straightforward. Let us define a Kramers-Wannier duality transformation for clock variables[9] $\hat{Z}_v \hat{Z}_{v+1} = \hat{\mathcal{Z}}_e$, which relates the degrees of freedom on the edge $e$ and the two adjacent vertices $v$ and $v + 1$. As in the edge picture, we define the following states on vertices: $|0\rangle \equiv |\blacksquare\rangle$, $|1\rangle \equiv |\blacksquare\rangle$, $|2\rangle \equiv |\blacksquare\rangle$. With open boundary conditions we must add an extra spin in the vertex picture that can be chosen arbitrarily (making the mapping one-to-three). Setting the boundary spin to $|\blacksquare\rangle$, the duality acts as

$$|\phi\,\phi\,\phi\,\phi\,\phi\,\phi\,\phi\,\phi\rangle \mapsto |\blacksquare\,\blacksquare\,\blacksquare\,\blacksquare\,\blacksquare\,\blacksquare\,\blacksquare\,\blacksquare\,\blacksquare\,\blacksquare\rangle . \tag{25}$$

The flippable pairs in the edge picture have been replaced by motifs of the form $|\blacksquare\,\blacksquare\,\blacksquare\rangle$ or $|\blacksquare\,\blacksquare\,\blacksquare\rangle$, i.e., a central spin surrounded by two spins of equal value. If the pair from which this motif is derived is permuted, $00 \to 11 \to 22 \to 00$, then, in the dual picture, this has the effect of permuting the central spin only, $0 \to 1 \to 2 \to 0$, leaving its neighbors unchanged. Hence, in the dual language, the generic pair-flip model from Sec. 2 may be written

$$\hat{H}_{\mathrm{PF}} = g \sum_{v=1}^{L-1} \sum_{\alpha,\beta,\gamma=0}^{m-1} \xi_v^{\alpha+\gamma,\beta+\gamma} \, \hat{P}_{v-1}^\gamma \hat{X}_v^{\alpha-\beta} \hat{P}_v^\beta \hat{P}_{v+1}^\gamma + \mathrm{h.c.}. \tag{26}$$

Note that the central projector does not affect the possible transitions; it merely ensures that the appropriate matrix element $\xi_v^{\alpha\beta}$ is selected. In the above equations $\hat{X}|\alpha\rangle = |\alpha + 1\rangle$ permutes the dual $m$-level systems, while $\hat{P}_v^\alpha = |\alpha\rangle\langle\alpha|_v$ projects onto color $\alpha$ on vertex $v$.

As stated above, in the dual language (26), spins are only dynamical if their neighbors are equal to one another. This model strongly resembles the PXP model [8, 61–64], for spin-1/2 degrees of freedom, where spins may only flip if both neighbors are in the '0' (ground) state due to the Rydberg blockade constraint. Equation (26), on the other

---

[9]This transformation differs slightly from the "standard" Kramers-Wannier transformation for clock variables, which takes the form $\hat{\mathcal{X}}_e = \hat{Z}_v^\dagger \hat{Z}_{v+1}$.

hand, also permits the central spin to flip if both neighbors are in the '1' (excited) state. The dual model (26) therefore generalizes the Rydberg constraint to a *parity*-sensitive constraint in which only the parity of neighboring 1's determines whether a given spin can flip.

The model in the vertex language appears to have an extra discrete symmetry

$$\prod_v \hat{X}_v^{(-1)^v} = \hat{X}_0 \hat{X}_1^\dagger \hat{X}_2 \hat{X}_3^\dagger \cdots ,$$

due to the choice of initial spin state. This symmetry can be retained in the edge picture by keeping track of an extra degree of freedom in the edge picture as a noninteracting spin on which the symmetry acts. Then, both pictures enjoy a $\mathbb{Z}_m$ symmetry, although it is local in the edge picture and global in the vertex picture. This is a generic feature of nonlocal dualities [65].

In the parity-sensitive PXP picture, we can also create Krylov sector labels via (i) working from left to right, identify motifs of the form $|\blacksquare \; \blacksquare \; \blacksquare\rangle$ (including three identically colored sites), (ii) for each motif identified, replace the pattern by the majority color, e.g., $|\blacksquare \; \blacksquare \; \blacksquare\rangle \mapsto |\blacksquare\rangle$, (iii) repeat the previous two steps until there are no such motifs remaining. This prescription leads to a dual conserved pattern. As an example, all of the following configurations [which are connected by local dynamics generated by the dual Hamiltonian (26)] map to the same conserved pattern:

$$\left[\, |\blacksquare \; \blacksquare \; \blacksquare \; \blacksquare\rangle \leftrightarrow |\blacksquare \; \blacksquare \; \blacksquare \; \blacksquare \; \blacksquare\rangle \leftrightarrow |\blacksquare \; \blacksquare \; \blacksquare \; \blacksquare\rangle \,\right] \mapsto |\blacksquare \; \blacksquare\rangle \, . \tag{27}$$

In general it is not necessary to start with the left-most motif; all choices lead to the same label. Note that the patterns on the left-hand side are dual to the patterns in (5) and the label on the right-hand side is dual to the label $|\blacklozenge\rangle$ that would be found from removing paired spins from (5).

### 4.2.2   Two-dimensional generalization

As for the one-dimensional model, we may introduce dual variables living on vertices via a similar Kramers-Wannier duality. These variables satisfy $\hat{Z}_v \hat{Z}_{v'} = \hat{\mathcal{Z}}_e$ for the two neighboring vertices $\langle vv'\rangle$ at either end of the edge $e$. Note that, for $i$, $j$, $k$, $l$ labeling the edges clockwise around a face $f$, we must have $\hat{\mathcal{Z}}_i \hat{\mathcal{Z}}_j^\dagger \hat{\mathcal{Z}}_k \hat{\mathcal{Z}}_l^\dagger = \mathbb{1}$ for $\hat{Z}_v$ to be independent of the path used to define it. This constraint is automatically satisfied within the source-free subspace.

We can therefore find (via a one-to-$m$ map) the spin configuration on vertices that corresponds to a given "domain wall" configuration on edges. For example,

$$\tag{28}$$

where we have shown the three possible states on the right-hand side, and spins on vertices have been represented using colored squares. The allowed configurations (9)

become

$$(29)$$

where $a + b = \alpha \mod m$ and $a + c = \beta \mod m$. More generally, if $i$, $j$, $k$, $l$ label spins on vertices clockwise around the face $f$, then the operator $\hat{N}^\alpha_{\partial f}$ that measures violations of the constraint, written in terms of the operators on vertices, is

$$\hat{N}^\alpha_{\partial f} = \frac{1}{m} \sum_{\beta=0}^{m-1} e^{-2\pi i \alpha\beta/m} (\hat{Z}^\beta_i - \hat{Z}^\beta_k)(\hat{Z}^\beta_j - \hat{Z}^\beta_l)\,, \tag{30}$$

and the ground space is still given by the requirement that $\hat{N}^\alpha_{\partial f} = 0$ for every face and color $\alpha$. From (29) or (30) it is clear that, in the vertex language, every face must have at least one identically colored diagonal pair in the source-free subspace. This suggests a further dual description of the system in terms of domain wall variables between such diagonal pairs [35]. More precisely, we introduce two degrees of freedom for every elementary face of the form $\hat{Z}^\dagger_i \hat{Z}_k$ and $\hat{Z}^\dagger_j \hat{Z}_l$, which correspond to domain wall variables between spins belonging to the same sublattice (i.e., even or odd). In this picture, the constraint implies that domain walls (defined by $\bar{Z}_{v_1} Z_{v_2} \neq 1$) between the even and odd sublattices cannot intersect, as illustrated in Fig. 4(c). A discussion of this final dual description is presented in Appendix A.

In the vertex language, the "parent" Hamiltonian is

$$\hat{H} = \tilde{J} \sum_f \sum_\alpha (N^\alpha_{\partial f})^2 - g \sum_v \sum_{\alpha=0}^{m-1} \sum_{\beta=0}^{m-1} \tilde{\xi}^{\alpha\beta}_v \, |\alpha\rangle\langle\beta|_e \,, \tag{31}$$

where $\tilde{J} > 0$, and the second term is the most general single-site term, which is assumed to be weaker than the first term. We can play the same game of projecting the dynamics into the constrained subspace, the groundspace of the first term. In this basis we only need to go to first order in perturbation theory, finding

$$\hat{H}_{\mathrm{QF}} = -g \sum_v \sum_{\alpha=0}^{m-1} \sum_{\beta=0}^{m-1} \tilde{\xi}^{\alpha\beta}_v \, |\alpha\rangle\langle\beta|_e \, \hat{\Pi}^\gamma_v, \tag{32}$$

where $\hat{\Pi}^\gamma_v = \prod_{v':\langle vv'\rangle} \hat{P}^\gamma_{v'}$ projects all vertices $v'$ that neighbor $v$ into the state $\gamma$. We can compare to Eq. (17) to see that $\tilde{\xi}^{\alpha\beta}_v \sim \xi^{\alpha+\gamma,\beta+\gamma}_v$. The QF Hamiltonian (17) is therefore dual to a Hamiltonian in which a spin is only dynamical if all four spins on its neighboring vertices are equal to one another. This correspondence is illustrated in Fig. 4, where flippable closed loops of length four in (a) map to vertices whose neighbors are all equal in (b). The model (32) is equivalent (for $m = 2$) to the first-order term that arises when projecting a transverse field into the ground space of the "$\mathsf{CZ}_p$" Ising model introduced in Ref. [35]. As in 1D, one can view Eq. (32) for $m = 2$ as arising

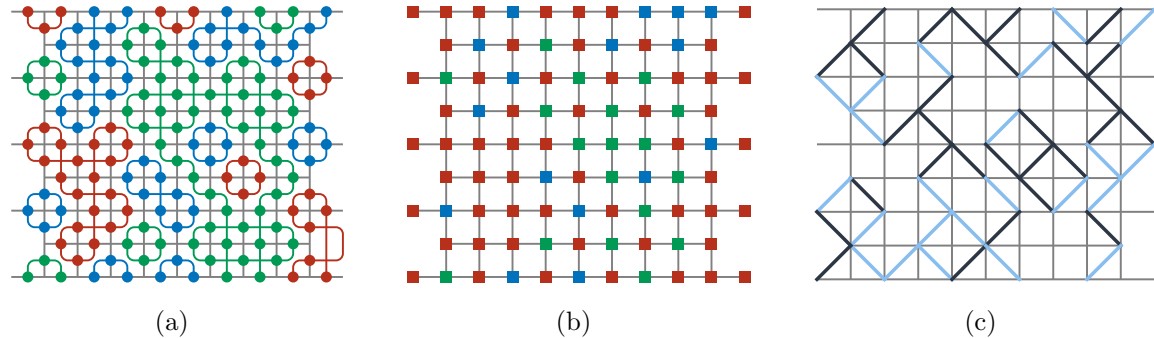

Figure 4: Three dual descriptions of the same state obeying the source-free constraint in models exhibiting topologically robust fragmentation. (a) The QF model (17), with degrees of freedom on edges. The four spins around a vertex are able to flip only if they match. (b) The parity-sensitive PXP model (32), with degrees of freedom on vertices. A given spin is able to flip only if its four neighbors are equal to one another. (c) The dual description introduced in Appendix A, where a domain wall is drawn if two adjacent spins belonging to the same sublattice do not match.

from a parity-sensitive Rydberg blockade constraint. Namely, as shown in Ref. [35], Eq. (32) arises at first order in a model where an odd number of neighboring 1's are forbidden around every face.

In fact, the $m = 2$ case of the 2D parity-sensitive PXP model with periodic boundary conditions is precisely the $\mathsf{CZ}_p$ model of Ref. [35]. The $\mathsf{CZ}_p$ model exhibits topologically robust ergodicity breaking, but only in maximal one-form charge sectors and with those sectors being fully frozen. Maximal charge sectors shatter into $\mathcal{O}(2^L)$ frozen states, while nonmaximal sectors are fully ergodic. Meanwhile, with open boundary conditions, the maximal charge sectors of the $\mathsf{CZ}_p$ model have only a single state and there is no ergodicity breaking to be had. In contrast, with $m > 2$ the parity-sensitive PXP model (like the QF model) exhibits ergodicity breaking with both periodic and open boundary conditions, there is fragmentation in generic symmetry sectors (not just sectors of maximal charge), and the fragmentation is not 'all or nothing,' i.e., there exist Krylov subsectors with a distribution of sizes between one (frozen states) and exponentially large in system size. the size of the full symmetry sector. As with the $\mathsf{CZ}_p$ model, this basic phenomenology is topologically robust.

## 4.3    Three-dimensional models

The natural generalization of the PF and QF models to 3D is a model on a cubic lattice with a 2-form symmetry. To define the model, let us put $m$-level degrees of freedom (1) on the edges of a cubic lattice. The symmetry operators take the same form as in Eq. (8), now defined on one-dimensional paths embedded in three-dimensional space. The 2-form symmetry once again breaks the constrained Hilbert space into $\mathcal{O}(L^{m-1})$ symmetry sectors.

Requiring $\hat{N}_{\partial f}^{\alpha} = 0$ on all faces $f$ allows the same configurations as shown in Eq. (9), now on faces in any orientation. Graphically, these constraints require that the 3D state decomposes into a collection of closed membranes that do not intersect, both contractible and noncontractible.

The allowed dynamics involve flipping all the spins around a vertex, of which there are now six. We will call the whole family of models the vertex-flip (VF) models. When viewed on a 2D slice of the system, the allowed moves look the same as those in (11). The VF move changes the color of the smallest contractible membrane:

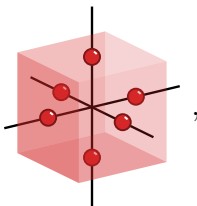

,

where the red spheres live on the edges of the primary cubic lattice and the shaded faces connect vertices of the dual cubic lattice. As in 2D, any contractible membrane may be flipped by a series of VF moves. Flipping a noncontractible membrane without violating the constraint now requires simultaneously flipping $\mathcal{O}(L^2)$ spins.

Operators charged under the 2-form symmetry are membranes on the dual lattice. As in 2D, they have nontrivial microscopic geometric requirements. They are required to turn at every edge of the dual lattice. For example, the operator

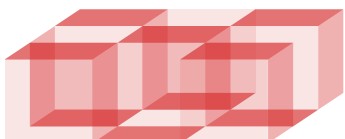

is a section of a valid charged operator, drawn on the dual lattice. For clarity of illustration, we have omitted the spins and the edges of the underlying lattice.

We can once again define rough and smooth boundary conditions. The smooth boundary conditions are simple: just truncate the lattice so that no edges stick out. Charged operators may end on this boundary but symmetry operators may not. The rough boundaries must again be defined on a staggered lattice truncation. In a cubic lattice, the edges sticking out from a ordinary rough boundary form a square lattice. Now, remove the outward-pointing edges from one sublattice. This leaves behind doubled boundary faces, on which we permit configurations as in (13). Finally, we put the rough boundary conditions on two opposite boundaries and smooth boundary conditions on the other four boundaries. With this choice of boundary conditions, the counting of sectors works the same way as in the PF and QF models. There are $\mathcal{O}(L^{m-1})$ symmetry sectors and $\mathcal{O}[(m-1)^L]$ Krylov sectors, some frozen and some nonfrozen.

Sectors are labeled by a pattern of system-spanning membranes of nonrepeating color. One key difference to the two-dimensional case is that, if all source-free states

have equal energy, there is a macroscopic energy barrier between states from any two different Krylov sectors. If, instead, source-free states are separated by an extensive energy $\propto g$, there can exist energetic competition between the constraint violations (scaling as $J|\partial R|$) and this extensive energy (scaling as $g|R|$), leading to an energy barrier $\sim J^2/g$ that diverges as $g \to 0$. This timescale is analogous to magnetization reversal in a 2D Ising model at finite temperature in the presence of a longitudinal field. This energy barrier could make the fragmentation in the 3D VF model even more robust than the fragmentation in the QF model. This possibility remains to be fully explored.

# 5   Conclusions

We have constructed a family of quantum loop based models exhibiting ergodicity breaking with topological robustness. These models simultaneously generalize the $CZ_p$ model of Ref. [35] to local Hilbert space dimensions $m > 2$ (in a dual representation), and the pair-flip model [41, 42] to higher dimensions, with the extra dimension(s) endowing the system with a topological stability that the pair-flip model lacks. The ergodicity breaking derives from $(d-1)$-form symmetries in $d > 1$ spatial dimensions, which can emerge (in a prethermal sense) from simple parent Hamiltonians that softly (i.e., energetically) implement a 'source-free' constraint. The simplest $m = 2$ version of this physics—the $CZ_p$ model—exhibits exponential fragmentation of Hilbert space, but only with periodic boundary conditions, and only in sectors of extremal symmetry charge. Moreover, the fragmentation is 'all or nothing,' i.e., either a given symmetry sector is fully frozen, or it is not fragmented at all. Once we generalize to $m > 2$, fragmentation is no longer limited to sectors of extremal charge, survives (suitable) open boundary conditions, and is not 'all or nothing.' That is, there arise Krylov subsectors of size intermediate between one (frozen states) and exponentially large in system size. All of this phenomenology is topologically stable, i.e., it is robust to arbitrary $k$-local perturbations, as long as $k/L \to 0$ as we take system size $L \to \infty$. The results are exact if the constraint is imposed exactly (exact one-form symmetry), and valid up to an exponentially long prethermal timescale if the constraint is imposed energetically (emergent one-form symmetry).

This work opens up a new direction for exploration of ergodicity-breaking quantum dynamics. The models we have constructed provide proof of principle that qualitatively new phenomena in many-body quantum dynamics can arise protected by higher-form symmetries (which can emerge from simple local Hamiltonians). The exploration of emergent higher-form symmetries and their consequences, however, has only just begun. Particularly fruitful in this regard is the extension of such constructions to three spatial dimensions – we have sketched some considerations in Sec. 4.3, but a detailed exploration remains to be performed. We look forward to further exploration of these ideas.

It is also interesting to note that our results are exact when the higher-form symmetries are exact, and it has been shown that emergent higher-form symmetries can in fact be exact in the low-energy subspace [39, 66]. Might results analogous to ours

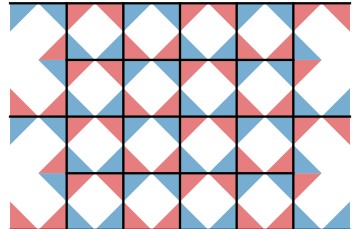

Figure 5: The sign convention that we introduce to define domain wall variables of the form $\hat{Z}_i^\dagger Z_j$ living on faces, which correspond to domain walls amongst even sublattice spins and amongst odd sublattice spins. Vertices that contribute a $Z$ ($Z^\dagger$) to the domain wall degree of freedom on a given face are denoted by red (blue) shaded regions. This choice affects the local constraints that any domain wall configuration must satisfy.

therefore continue to hold even beyond the exponentially long 'prethermal' timescale that we estimate? This too presents a fruitful topic for future investigation.

Finally, while our discussion has focused on the quantum *dynamics*, the equilibrium properties of generalized loop model could also be of interest. During the preparation of this manuscript, we became aware of parallel work by Shankar Balasubramanian, Ethan Lake, and Soonwon Choi [67] on the ground state properties of no-crossing models, and this too presents an interesting direction for future work.

### Acknowledgements

RN and OH would like to thank David T. Stephen for a prior collaboration on the $\mathsf{CZ}_p$ model. This work is supported by the Air Force Office of Scientific Research under award number FA9550-20-1-0222.

## A   Additional dualities

Here, we describe a further duality of the two-dimensional models (17) and (32), which involves introducing two degrees of freedom per face that correspond to domain wall variables between spins belonging to the same sublattice. That is, a spin situated at a vertex with coordinates $(x, y)$ belongs to the even (odd) sublattice if $x + y$ is even (odd). The introduction of these new degrees of freedom is motivated by Eq. (30), which requires that, around a given face, the two spins on the even sublattice must match, or the two spins on the odd sublattice must match (or both). Explicitly, given four vertices around a face, $v_1$, $v_2$, $v_3$, and $v_4$ (no ordering implied), with $v_1, v_3$ belonging to the even sublattice, and $v_2, v_4$ to the odd sublattice, we construct domain wall variables $\hat{\tau}_{f,+}^z = \hat{Z}_{v_1}^\dagger \hat{Z}_{v_3}$ and $\hat{\tau}_{f,-}^z = \hat{Z}_{v_2}^\dagger \hat{Z}_{v_4}$. In this language, the constraint is particularly simple: On any face $f$, domain walls (defined by $\tau_{f,s}^z \neq 1$) cannot intersect.

For $m > 2$, the $\hat{Z}_v$ operators are not Hermitian, so there exists a choice in which $\hat{Z}$ operators are conjugated. Illustrating those vertices that contribute a $\hat{Z}$ ($\hat{Z}^\dagger$) by red

(blue) shaded regions, the convention that we utilize is illustrated in Fig. 5. In principle, there are $m - 1$ species of domain wall per sublattice. However, when illustrating spin configurations we choose not to distinguish between domain walls on the even and odd sublattices. The configurations on faces for $m = 3$ consistent with the constraint are:

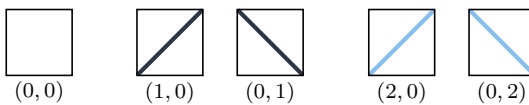

where the labels denote $(\alpha_{f,+}, \alpha_{f,-})$ assuming the top left and bottom right vertices belong to the even sublattice (for a given face and sublattice, $\alpha$ is defined by $\tau^z = e^{2\pi i \alpha/m}$). The convention in Fig. 5 enforces constraints on the colors of domain walls on adjacent faces. Namely, the domain wall variables on the four faces around a vertex $v$, belonging to sublattice $s$, satisfy $\prod_{f \in v} \hat{\tau}^z_{f,-s} = \mathbb{1}$, equivalent to $\sum_{f \in v} \alpha_{f,-s} = 0 \mod m$. Graphically, a domain wall will therefore change color at every vertex, unless it branches or meets another domain wall (at a vertex), as shown in Fig. 4(c).

# B   Enumerating sectors and frozen states

## B.1   Krylov sectors in the pair-flip model

The Bethe lattice mapping presented in Fig. 1 can also be applied in 1D to facilitate the counting of labels and Krylov sector dimensions [41, 42]. The 'final' position on the Bethe lattice after traversing the entire system from left to right, having begun at the root node of the Bethe lattice, is in one-to-one correspondence with the label introduced in Sec. 2, which identifies distinct Krylov sectors in the 1D pair-flip model with open boundaries. Similarly, each walk is in one-to-one correspondence with a spin configuration. Hence, the number of spin configurations that map to the same label (with length $\ell$) is equal to the number of walks of length $L$ that reach a given point with depth $\ell$ on the Bethe lattice (note that this number will depend only on $\ell$). The number of such walks $G_L^{(\ell)}$ – equal to the dimension of the Krylov sector for a label of length $\ell$ – is enumerated by the generating functions found in Refs. [41, 68]:

$$G^{(\ell)}(x) = \sum_{L=0}^{\infty} G_L^{(\ell)} x^L = \left( \frac{1 - \sqrt{1 - 4(m-1)x^2}}{2(m-1)x} \right)^\ell \frac{2(m-1)}{m - 2 + m\sqrt{1 - 4(m-1)x^2}} . \quad (33)$$

Note that $G^{(\ell)}(x)$ enforces that $G_L^{(\ell)} = 0$ if $\ell$ and $L$ have opposite parity, as is required. The number of sites on the Bethe lattice that are accessible after $L$ steps is given by Eq. (7). The exponential growth the Krylov sector dimension $G_L^{(\ell)}$ is determined directly (up to subexponential factors) by the radius of convergence of Eq. (33) [69]. Specifically, we have

$$G_L^{(\ell)} \sim (2\sqrt{m-1})^L \quad (34)$$

as $L \to \infty$ for fixed $\ell$. Insight into the subexponential factors can be obtained by performing a singularity analysis of Eq. (33) [41, 69, 70]. Expanding around the singularities at $x = \pm 2\sqrt{m-1}$, we find that

$$G_L^{(\ell)} = 2\left(\ell + \frac{m}{m-2}\right)\frac{m-1}{m-2}\sqrt{\frac{2}{\pi L^3}}2^L\sqrt{m-1}^{L-\ell}\left[1 + \mathcal{O}(L^{-1})\right] , \qquad (35)$$

as $L \to \infty$ for fixed, $\mathcal{O}(1)$ values of $\ell$. This result shows that, at least for $m > 2$, sectors with larger label lengths $\ell$ are exponentially suppressed in size with respect to the $\ell = 0$ sector. Additionally, we can deduce that no Krylov sector grows faster than $(2\sqrt{m-1})^L$ as $L \to \infty$.

We can also write down a generating function that counts the number of labels belonging to each *symmetry sector*. That is, using this generating function, we can deduce into how many Krylov sectors a particular symmetry sector decomposes. To do this, we must account for the U(1) charges associated with each of the $m$ colors. Let the generating variables $\mathbf{y} = \{y_\alpha\}$ keep track of the charge $N^\alpha$ defined in Eq. (2). Restricting to $m = 3$ for simplicity, we can then introduce the two transfer matrices [the row (column) index corresponds to the current (previous) step]

$$T_\sigma = x\begin{pmatrix} 0 & y_r^\sigma & y_r^\sigma \\ y_g^\sigma & 0 & y_g^\sigma \\ y_b^\sigma & y_b^\sigma & 0 \end{pmatrix} , \qquad (36)$$

with $\sigma = \pm 1$. More generally, the transfer matrix will be an $m \times m$ matrix with the above structure; the zeros along the diagonal enforce that the color of a given dot cannot match the color of the previous dot in the label. The sign $\sigma$ corresponds to whether an even or an odd edge is being traversed, which add to or subtract from the corresponding U(1) charge. The generating variable $x$ records the length of the label. For open boundary conditions, the full generating function admits the expansion

$$F(x; \mathbf{y}) = 1 + \sum_{i=1}^{m}\left(T_0 + T_-T_0 + T_+T_-T_0 + T_-T_+T_-T_0 + \dots\right)_i , \qquad (37)$$

where the initial condition $T_0 = x(y_r, y_g, y_b)^T$ corresponds to the first (unconstrained) dot, which we assume belongs to the even sublattice. In the presence of periodic boundaries, an analogous expression can be obtained by removing the initial condition $T_0$ and replacing the sum over $i$ by a trace. Factoring out the repeating combination $T_+T_-$, we find that

$$F(x; \mathbf{y}) = 1 + \sum_{i=1}^{m}\sum_{\ell=1}^{\infty}[(\mathbb{1} + T_-)(T_+T_-)^{\ell-1}T_0]_i \qquad (38a)$$

$$= 1 + \sum_{i=1}^{m}[(\mathbb{1} + T_-)(\mathbb{1} - T_+T_-)^{-1}T_0]_i . \qquad (38b)$$

where we used $\sum_{n=0}^{\infty} A^n = (\mathbb{1} - A)^{-1}$, which is convergent if the spectral radius of the matrix $A$ is strictly less than unity. The matrix inverse in Eq. (38b) can be evaluated exactly to arrive at the expression

$$F(x; \mathbf{y}) = \frac{(1 - x^2) \left[1 + x(y_r + y_g + y_b) + 2x^2\right]}{1 + x^2 + 4x^4 - x^2 \left[\frac{y_r}{y_g} + \frac{y_r}{y_b} + \frac{y_g}{y_b} + \frac{y_g}{y_r} + \frac{y_b}{y_r} + \frac{y_b}{y_g}\right]} . \tag{39}$$

As required, throwing away the information about the U(1) charges reduces to $F(x; \mathbf{1}) = (1 + x)/(1 - 2x)$, with coefficients $3 \times 2^{n-1}$ for $n \geq 1$, equal to the total number of labels of length $n$. Performing the same procedure for case $m = 2$ provides another simple example:

$$\frac{y_r y_b (1 - x^2) \left[1 + x(y_r + y_b) + x^2\right]}{(x^2 y_r - y_b)(x^2 y_b - y_r)} = 1 + \sum_{n=1}^{\infty} \frac{y_r^n + y_b^n}{(y_r y_b)^{\lfloor n/2 \rfloor}} x^n , \tag{40}$$

since there is only one label associated with each symmetry sector.

Returning to $m = 3$, let us evaluate the total number of labels that live in the symmetry sector with vanishing charge for all three colors, $N^\alpha = 0$. Such "uncharged" labels are also useful for constructing labels of arbitrary length compatible with a given set of symmetry quantum numbers: Given a label belonging to the appropriate sector, an uncharged label can be appended or prepended (subject to the constraint that neighbors may not match) to produce new labels belonging to the same sector. To extract the generating function of uncharged labels, we are required to evaluate the integral

$$\int_{-\pi}^{\pi} \frac{\mathrm{d}\phi_r \mathrm{d}\phi_g \mathrm{d}\phi_b}{(2\pi)^3} \frac{(1 - x^2) \left[1 + x(e^{i\phi_r} + e^{i\phi_g} + e^{i\phi_b}) + 2x^2\right]}{1 + x^2 + 4x^4 - 2x^2 \left[\cos(\phi_r - \phi_g) + \cos(\phi_g - \phi_b) + \cos(\phi_b - \phi_r)\right]} . \tag{41}$$

The number of labels belonging to other symmetry sectors can be obtained similarly by first multiplying by $\exp(-i \sum_\alpha \phi_\alpha N^\alpha)$. We can make progress by shifting (say) $\phi_{g,b} \rightarrow \phi_{g,b} + \phi_r$, making the integral over $\phi_r$ trivial. Additionally, we define the function $E(x) = (1 + x^2 + 4x^4)/x^2$ and restructure the trigonometric functions, which brings the integral into the form

$$\frac{(1 - x^2)(1 + 2x^2)}{x^2} \int_{-\pi}^{\pi} \frac{\mathrm{d}k_1 \mathrm{d}k_2}{(2\pi)^2} \frac{1}{E(x) - 2\left[\cos(k_1) + 2\cos(\frac{k_1}{2})\cos(\frac{k_1}{2} - k_2)\right]} . \tag{42}$$

This integral is equal to the Green's function of a single tight-binding quantum particle hopping on a triangular lattice [71], for which (in appropriate coordinates) the dispersion relation may be written $E(\mathbf{k}) = 2\left[\cos(k_1) + 2\cos(\frac{k_1}{2})\cos(\frac{k_1}{2} - k_2)\right]$. We can therefore write down an exact expression for the generating function of uncharged configurations

$$F_0(x) \equiv [y_r^0 y_g^0 y_b^0] F(x; \mathbf{y}) = \frac{2(1 - x^2)(1 + 2x^2)}{\pi x^2 (\lambda - 1)^{3/2}(\lambda + 3)^{1/2}} K\left(\frac{4\lambda^{1/2}}{(\lambda - 1)^{3/2}(\lambda + 3)^{1/2}}\right) , \tag{43}$$

where $K(x)$ is the complete elliptic integral of the first kind and we have introduced $\lambda(x) = \sqrt{3 + E(x)}$. The first few pattern lengths that belong to the uncharged symmetry sector that derive from Eq. (43) are

$$F_0(x) = 1 + 6x^6 + 6x^8 + 42x^{10} + 120x^{12} + 426x^{14} + \ldots \tag{44}$$

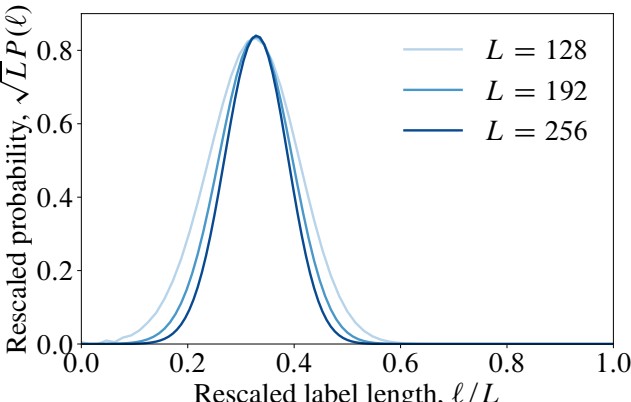

Figure 6: Exact distribution of Krylov sectors in the uncharged symmetry sector, obtained from the generating functions (33) and (43). $P(\ell)$ is the probability that a state drawn at random from the uncharged symmetry sector with uniform probability belongs to a Krylov sector corresponding to label length $\ell$. The largest Krylov sector, which corresponds to label length $\ell = 0$, represents a vanishingly small fraction of the symmetry sector.

The shortest uncharged pattern has length six and corresponds to a three-dot pattern repeated twice, such as $|\!\!\,\bullet\,\bullet\,\bullet\,\bullet\,\bullet\,\bullet\rangle$. The six labels of length $\ell = 8$ are the same six labels as $\ell = 6$ surrounded by the unique color that is not equal to the first or last color in the corresponding $\ell = 6$ label. Asymptotically, the number of patterns of length $\ell$ within the uncharged symmetry sector scales (up to polynomial corrections) as $\sim 2^\ell$.

Lastly, we note that fragmentation in the PF model is *strong* [11, 12]: In the thermodynamic limit, an arbitrary state chosen from any fixed symmetry sector (i.e., one whose charge does not scale with $L$) belongs to that sector's largest Krylov sector with probability zero. To see this, recall that any symmetry sector has a minimal length $L_{\min}$ on which it exists. To bound the size of the symmetry sector, we can then place any uncharged spin configuration on the remainder of the system. For $m = 3$ and systems of length $L = L_{\min} + 6n$, we can place $n$ red spins, $n$ blue spins, and $n$ green spins on the even sublattice (and similarly for the odd sublattice), giving

$$D_0^{\mathrm{sym}} > \left[ \frac{(3n)!}{(n!)^3} \right]^2 \sim \frac{3^{6n+1}}{4\pi^2 n^2} \tag{45}$$

uncharged spin configurations. Hence, every fixed symmetry sector has a dimension that scales asymptotically as $\sim 3^{L-L_{\min}}$ up to polynomial corrections, while no Krylov sector grows faster than $\sim (2\sqrt{2})^L$ (34). For larger $m$, symmetry sectors contain $\sim m^L$ states, while no Krylov sector grows faster than $\sim (2\sqrt{m-1})^L$. This result is illustrated for the uncharged symmetry sector in Fig. 6, which shows that an arbitrary state is most likely to belong to a Krylov sector of intermediate size, and that the probability of belonging to any particular Krylov sector vanishes in the thermodynamic limit.

## B.2   Labels in 1D with periodic boundaries

Let us consider PF dynamics with periodic boundaries. The procedure for finding the label is the same, except that spins on the left and right ends of the system may be paired so that the first and last spin in the label must not match. For labels of size $L$ this is the end of the story; there are $(m-1)^L + (-1)^L(m-1)$ such sectors, the number of $m$-colorings of the cycle $C_L$. Each sector consists of a single frozen state.

For shorter labels, the dots are mobile, in the sense of Eq. (5). This allows us to cyclicly translate the label by a distance of two. Instead of just keeping track of the length of the label $\ell$, let us also keep track of the shortest repeating pattern ("motif") in a label and its length $j$. A motif can be repeated up to $n = \lfloor L/j \rfloor$ times. For $j$ odd, a nonfrozen sector is labeled by a motif and a choice of $n$. For $j$ even (which can only occur if $L$ is even), a nonfrozen sector is labeled by a motif, $n$, and a choice of parity bit, since the label can only be shifted two positions at a time. Note that we could extend this labeling to frozen states if we supplement with a starting position within the motif, of which there are $j$.

The number of motifs of length $j$ is given by the recurrence relation

$$N_j^{\text{motif}} = (m-1)^j + (-1)^j(m-1) - \sum_{k\,|\,j} N_k^{\text{motif}}, \tag{46}$$

where $k\,|\,j$ means $k$ divides $j$. The subtraction removes motifs that are periodic with period $k$. Asymptotically, we have $N_j^{\text{motif}} \sim (m-1)^j$. Explicit expressions for the subleading corrections can be found when $j$ is a power of a prime, but are not particularly illuminating. Then, the number of labels of length $\ell$ is dominated by labels with a single motif, and also scales as $(m-1)^\ell$. But, since the dots are mobile, sectors correspond to labels up to translation by 2. For odd $j$, translations by 2 are fully general so that any two motifs related by translation correspond to the same sector when repeated $n$ times. This tells us there are $\sim (m-1)^\ell/\ell$ sectors with label of length $\ell$. If $j$ is even then two motifs that are related by a translation by one cannot be transformed into each other. This means that sectors must have an additional parity bit which introduces a factor of 2 into the counting, but does not affect the scaling.

This all tells us that despite having $\sim (m-1)^L$ frozen states, the PF model with periodic boundary conditions asymptotically has only $\sim (m-1)^L/L$ nonfrozen sectors. This contrasts with open boundary conditions, where we found $\sim (m-1)^L$ frozen sectors and $\sim (m-1)^L$ nonfrozen sectors.

## B.3   Labels in 2D with periodic boundaries

The labeling procedure in 2D with PBC is more complicated still, but does have some nice graphical interpretation. We now have the option of choosing a motif with length $j$ and two integers, $n_x$ and $x_y$, such that $\max(|jn_x|, |jn_y|) \leq L$. Then the horizontal label is the motif repeated $n_x$ times (from left to right) and the vertical label is the motif repeated $n_y$ times (from top to bottom). Negative values correspond to repeating the

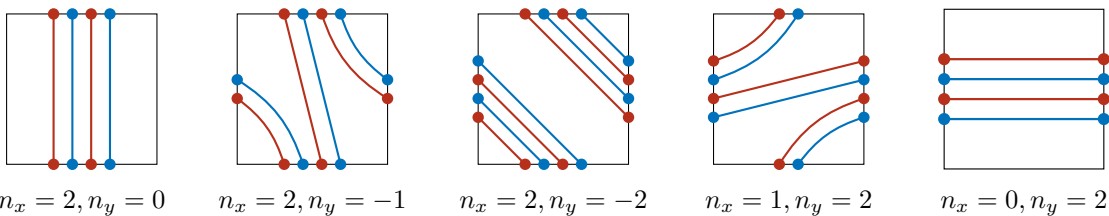

$$n_x = 2, n_y = 0 \qquad n_x = 2, n_y = -1 \qquad n_x = 2, n_y = -2 \qquad n_x = 1, n_y = 2 \qquad n_x = 0, n_y = 2$$

Figure 7: In 2D, we can choose to repeat a particular motif (in this case, $|\!\!\diamond\!\!\diamond\rangle$) an integer number of times both horizontally and vertically. For example, in the left-most figure the motif is repeated $n_x = 2$ times horizontally and $n_y = 0$ times vertically. In the middle figure we have $n_y = -2$ because the label is reversed when read from top to bottom. We could in addition have states with $n_x > 2$ or $n_y > 2$, with an upper limit set by $L/2$ ($L/j$ in general).

motif in reverse. A sector consists of a single frozen state if either $jn_x = L$ or $jn_y = L$. Such states are discussed in Sec. B.4. Here, we will focus on nonfrozen sectors.

Different choices of $n_x$ and $n_y$ define how many times a motif is repeated horizontally and vertically. These values also define the average slope of the noncontractible loops: The average slope is $\pm n_x / n_y$. Figure 7 demonstrates how a particular motif can be included different numbers of times either horizontally or vertically.

As in 1D, two nonmaximal labels related by translation (by two) belong in the same sector. This means there are once again only $\sim (m-1)^L / L$ nonfrozen sectors in PBC, compared to $\sim (m-1)^L$ nonfrozen sectors in OBC, or $\sim (m-1)^L$ frozen states in either case.

## B.4   Frozen states in the QF model

The counting of frozen states in the presence of periodic boundaries is more complicated than the case of open boundary conditions presented in the main text. We will work using the language of the QF Hamiltonian (i.e., spins on edges) and with square systems of size $L \times L$ for simplicity. The first ingredient in counting the number of such states is to identify the number of 1D configurations that are not mobile under pair-flip dynamics; namely, the number of configurations that do not contain any neighbors of the same color. When these 1D configurations are turned into system-winding loops in the second dimension, there will be no adjacent loops of the same color. For a ring of length $j$, the number of configurations with no identically colored neighbors is $(m-1)^j + (m-1)(-1)^j$. Suppose that such a constraint-satisfying pattern with $j = L$ is placed in the first row of the system. In the subsequent rows, the pattern can be shifted either left or right subject to the constraint that it must come back to itself around the periodic boundaries. Consequently, any *periodicity* of the pattern plays a nontrivial role; if the pattern repeats every $j$ edges, the final displacement $x$ of the pattern need only satisfy $x = 0 \mod j$.

For simplicity, we will first focus on linear system sizes of the form $L_k = 2^k$, although the methods we present can be used to identify the number of frozen states for arbitrary $L$. Given this simplification, the pattern can, in principle, repeat every $j_n = 2^n$ edges for

$1 \leq n \leq k$. For $k > 1$, the number of frozen patterns of length $L_k$ with no periodicity is

$$N_k = (m-1)^{L_k} + (m-1)(-1)^{L_k} - \sum_{1 < n < k} N_n \,, \tag{47}$$

where the second term on the right-hand side subtracts off the contribution from periodic patterns; more generally, one must sum over all divisors of the length of the pattern, as in Eq. (46). The recursion relation (47) can be solved exactly to give

$$N_k = (m-1)^{L_k/2} \left[ (m-1)^{L_k/2} - 1 \right] \text{ for } k > 1 \,, \tag{48}$$

and $N_1 = m(m-1)$. Asymptotically, we have $N_k \sim (m-1)^{L_k}$ for $k \gg 1$, as expected. That is, the contribution from periodically repeating patterns is exponentially suppressed. For a system of size $L_k \times L_k$, the full number of frozen configurations that wrap around the system in (at least) one direction is therefore

$$F_k = \sum_{j \,|\, L_k} \sum_{n=0}^{2L_k/j} \binom{L_k}{\frac{1}{2}nj} N_\ell \,, \tag{49}$$

where the first summation is over nontrivial motif lengths $j$ that divide $L_k$, i.e., $j \in \{2^n\}_{n=1}^k$. The leading term in the summation comes from the term $j = L_k$, which gives (for $m > 2$) the asymptotic growth

$$F_k = \left[ 2 + \binom{L_k}{\frac{1}{2}L_k} \right] N_{L_k} + O\left( \frac{2^{L_k}}{\sqrt{L_k}} (m-1)^{L_k/2} \right) \sim \sqrt{\frac{2}{\pi L_k}} [2(m-1)]^{L_k} \,, \tag{50}$$

For $m = 2$, there is just a single frozen pattern composed of alternating colors of loops. Since this pattern necessarily has periodicity $j = 2$, there is no exponential enhancement of large, nonrepeating patterns. Hence, in this case, one must sum over all binomial coefficients, giving $\sum_{n=0}^{L_k} \binom{L_k}{n} = 2^{L_k}$. The asymptotic scaling in Eq. (50) is compared with the exact number of frozen states computed numerically for arbitrary $L$ in Fig. 8, which suggests that (50) describes the leading asymptotic growth for all even $L$. Note that an exact count of *all* frozen states would require us to enumerate configurations that wrap the torus in the other direction [not all of which are distinct from the states already counted in Eq. (49)]; the asymptotic growth, however, will remain unchanged.

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
