# Peer review of "Topologically stable ergodicity breaking from emergent higher-form symmetries in generalized quantum loop models"

_SciPost Physics_

## Round 2 · Referee Report · Anonymous (Referee 1) · 2023-7-9

Report

In this manuscript, the authors propose the quad-flip model which can be regarded as generalization of pair flip model in 2D dimension. By adding the constraint of free source, the system shows fragmentation of symmetry sectors, like the pair-flip model. Similarly, in the quad-flip model different Krylov sector can be labeled by the order of non-contractible paths. However, different from the pair-flip model, the fragmentation of Hilbert space is robust unless dynamics evolves degree with the order of system size. The authors also discuss the connection with other models such PXP model and the generalization to 3D case. I think this manuscript is technically correct and has high quality. I agree with publishing this manuscript.
  • validity: -
  • significance: -
  • originality: -
  • clarity: -
  • formatting: -
  • grammar: -

Author:  Charles Stahl  on 2023-12-28  [id 4213]

(in reply to Report 1 on 2023-07-09)

We thank the referee for the positive assessment of our work and for recommending publication in SciPost Physics.

---

## Round 2 · Referee Report · Anonymous (Referee 2) · 2023-8-9

Strengths

1- Clear presentation

2- Useful formatting and figures

3-Connects interesting and active topics (higher-form symmetries, ergodicity breaking)

Weaknesses

1- The novel parts of this work are mainly based on one specific model and it would be interesting to have a more generic/model-independent answer concerning the connection between ergodicity breaking and higher-form symmetries.

Report

This work presents models that exhibit topologically stable ergodicity breaking. This is accomplished by the generalization of [35], where one-form symmetry was already employed but the ergodicity breaking was "all or nothing", and the generalization of the pair-flip models [41,42] which are generalized to be topologically stable.

The introduction highlights the general question, its challenges, the state-of-the-art, and places the current work into context.

Section 2 is a useful warm-up that reviews the one-dimensional pair-flip model. The authors argue that the fragmentation crucially depends on the restriction to nearest-neighbor-dynamics (a feature that they want to remedy).

Section 3 generalizes the pair-flip model to two dimensions. Using a 1-form symmetry the fragmentation is robust under generic local perturbations (topologically robust) and not "all or nothing" anymore. This section provides the main results of this work.

Section 4 provides further, not strictly necessary, details concerning the two-dimensional model: Other theories which arise as limits or via dualities and a discussion of a generalization to three dimensions (where 2-form symmetries are relevant).

The authors provide a conclusion and two appendices with further details.

This work opens the pathway for the exploration of ergodicity-breaking quantum dynamics that derive from higher-form symmetries and also provides a synergetic link between these different research areas. I am therefore happy to recommend this work for publication in SciPost Physics.
  • validity: top
  • significance: top
  • originality: high
  • clarity: top
  • formatting: perfect
  • grammar: perfect

Author:  Charles Stahl  on 2023-12-28  [id 4212]

(in reply to Report 2 on 2023-08-09)

We thank the referee for the detailed and positive summary of our work, and for recommending publication.
The connection between ergodicity breaking and higher-form symmetries is actually rather subtle. For instance, in the class of square-lattice models we consider in this work, higher-form symmetry provides a powerful way of understanding the robust fragmentation. However, the same higher-form symmetry does not necessarily produce robust fragmentation on arbitrary lattices. Indeed, even formulating the symmetry on a lattice that is not bipartite is tricky. Resolving the question of how these models can be generalized to arbitrary lattices, and further clarifying the connections between higher-form symmetries and robust fragmentation, is an important and interesting problem for future work, but is beyond the scope of the current manuscript. We have added a paragraph to the conclusions discussing these subtleties.

---

## Round 2 · Referee Report · Anonymous (Referee 3) · 2023-8-28

Strengths

1- Novel results on a timely topic 2- Very clearly written manuscript

Report

The authors introduce a generalized quantum loop models for which they show topologically stable ergodicity breaking. The present work is a generalization of a recent paper by Stephen et., in which a robust ergodicity breaking is demonstrated. The ergodicity breaking in the new class of models is shown to be more robust in that it does not require dense packing nor periodic boundary conditions.

The results represent a novel mechanism allowing for a robust ergodicity breaking and are thus of strong interest. Moreover, the manuscript is well written and all results are presented clearly.

Minor comment: When reading the manuscript, I thought it might be useful to discuss whether the system is strongly or easily fragmented also in the main text.

In summary, this work contains novel results on a timely topic and the manuscript is very clearly written. I thus recommend publication in SciPost.
  • validity: top
  • significance: high
  • originality: high
  • clarity: top
  • formatting: perfect
  • grammar: perfect

Author:  Charles Stahl  on 2023-12-28  [id 4214]

(in reply to Report 3 on 2023-08-28)

We thank the referee for the favorable report and the recommendation for publication. To address the referee's one minor comment, we have added a new Appendix (Appendix C) discussing whether fragmentation in the model is weak or strong. To answer this question conclusively, we introduced a loop Monte Carlo algorithm for sampling from the constraint-satisfying subspace, which we used to show that the largest Krylov sector occupies almost all of Hilbert space in the thermodynamic limit, corresponding to weak fragmentation. We have also stated this result in the main text.

---

## Editorial Decision

resubmitted